# Mapping and Omics Integration: Towards Precise Rice Disease Resistance Breeding

**DOI:** 10.3390/plants13091205

**Published:** 2024-04-26

**Authors:** John Christian Ontoy, Jong Hyun Ham

**Affiliations:** 1Department of Plant Pathology and Crop Physiology, LSU AgCenter, Baton Rouge, LA 70803, USA; jontoy@agcenter.lsu.edu; 2Department of Plant Pathology and Crop Physiology, College of Agriculture, Louisiana State University, Baton Rouge, LA 70803, USA

**Keywords:** *Oryza sativa*, disease resistance, QTL mapping of rice disease resistance, multi-omics-driven rice breeding, integrative approach in studying rice disease resistance

## Abstract

Rice (*Oryza sativa*), as a staple crop feeding a significant portion of the global population, particularly in Asian countries, faces constant threats from various diseases jeopardizing global food security. A precise understanding of disease resistance mechanisms is crucial for developing resilient rice varieties. Traditional genetic mapping methods, such as QTL mapping, provide valuable insights into the genetic basis of diseases. However, the complex nature of rice diseases demands a holistic approach to gain an accurate knowledge of it. Omics technologies, including genomics, transcriptomics, proteomics, and metabolomics, enable a comprehensive analysis of biological molecules, uncovering intricate molecular interactions within the rice plant. The integration of various mapping techniques using multi-omics data has revolutionized our understanding of rice disease resistance. By overlaying genetic maps with high-throughput omics datasets, researchers can pinpoint specific genes, proteins, or metabolites associated with disease resistance. This integration enhances the precision of disease-related biomarkers with a better understanding of their functional roles in disease resistance. The improvement of rice breeding for disease resistance through this integration represents a significant stride in agricultural science because a better understanding of the molecular intricacies and interactions underlying disease resistance architecture leads to a more precise and efficient development of resilient and productive rice varieties. In this review, we explore how the integration of mapping and omics data can result in a transformative impact on rice breeding for enhancing disease resistance.

## 1. Introduction

Rice diseases are caused by fungi, bacteria, viruses, and other types of pathogens. Common diseases include blast, sheath blight, bacterial leaf blight, and tungro. Each disease has distinct symptoms and affects different parts of the rice plant, leading to yield losses if not managed effectively [1]. Understanding the intricate interactions between rice plants and pathogens is essential for developing more advanced and sophisticated disease management strategies, targeting specific host or pathogen elements important for susceptibility or resistance. Plant pathogens recognize specific signals from host plants, triggering the initiation of the infection process, and they employ diverse strategies to breach the plant’s physical barriers. Fungi, for example, produce enzymes to degrade plant cell walls, facilitating penetration. Bacteria use specialized secretion systems to deliver effector molecules into plant cells, modulating host responses and enabling invasion [2]. Secreted effector molecules (mostly proteins) manipulate host cellular processes and interfere with the defense responses of host plants, suppressing disease resistance. They may also mimic plant molecules to evade pathogen detection by the host plant. In other words, effectors play a central role in the establishment of infections by subverting plant immunity and creating a conducive environment for pathogen proliferation [3]. In response, plants have evolved sophisticated immune systems to counter pathogen attacks. Pattern recognition receptors (PRRs) detect conserved pathogen-associated molecular patterns (PAMPs), initiating PAMP-triggered immunity (PTI). Pathogens counter PTI with effectors, leading to effector-triggered immunity (ETI), a robust defense mechanism of plants. The balance between pathogen effectors and plant immune responses determines the outcome of the infection [4]. Rice possesses a variety of genes that confer resistance to specific pathogens. These genes, often referred to as R genes, produce proteins that recognize pathogen molecules, activating the plant’s defense mechanisms.

In general, there are two types of resistance in plants. While qualitative resistance is typically controlled by specific R genes and leads to a discrete resistant or susceptible outcome, quantitative resistance involves multiple genes and results in a spectrum of resistance levels. In rice, quantitative resistance is often more durable than qualitative resistance, as it involves a complex interplay of various genes, enhancing the plant’s ability to withstand diverse pathogen populations over time [5]. Combining quantitative resistance with major R genes has proven to be a valuable approach for extending the effectiveness of major genes or durability of resistance in rice [6]. In this article, we will discuss how we can achieve a holistic understanding of rice disease resistance through the integration of multi-omics approaches and, ultimately, translate it into the innovation of rice breeding for increasing disease resistance.

As technology continues to advance, the future holds promising prospects for even more precise and efficient methods in characterizing disease resistance mechanisms. As research progresses, the integration of mapping with other omics approaches is poised to further deepen our understanding of the genetic underpinnings of complex disease-related traits. Understanding the complex interactions between plants and pathogens at the molecular level is crucial for developing effective disease-resistant crops. Omics technologies, including genomics, transcriptomics, proteomics, and metabolomics, have revolutionized our ability to decode the intricate mechanisms underlying plant disease resistance in addition to other complex agronomic traits in rice breeding (Figure 1). Hence, we will briefly review how various omics studies have contributed to expanding our knowledge of plant disease resistance and how integrative multi-omics studies have been performed with other plant systems.

## 2. Omics for Decoding Resistance Mechanisms

### 2.1. Harnessing Genomics for Enhancing Rice Disease Resistance

Quantitative trait locus (QTL) linkage mapping is one of the methods for identifying genomic regions involving the construction of genetic maps that link phenotypic traits, such as disease resistance, to molecular markers or genes located on several or specific chromosomes. The breakthrough in the characterization of quantitative traits to select for QTLs was the development of molecular markers used for the construction of linkage maps for diverse crop species [7]. Linkage maps have been utilized for identifying chromosomal regions that contain genes controlling simple traits or quantitative traits governed by QTL [8]. Advantages of linkage mapping include high statistical power to effectively identify regions of genome associated with the target trait [9], trait specificity, which allows the mapping of complex traits like disease resistance [10,11], and the identification of linkage with functional variations such as genetic polymorphisms, gene expression changes, or epigenetic modifications, which can provide valuable information on the underlying mechanisms behind the trait variation [8]. Lastly, the method is suitable for diverse populations such as F_2_, backcross, and recombinant inbred lines (RILs) [8]. However, QTL studies necessitate large sample sizes due to their reliance on statistical power to detect small genetic effects and can only map differences observed between parents, as it is improbable for every genetic locus contributing to a variation to harbor segregating alleles of major effect within the populations [10,12]. Also, the limited genetic diversity within a biparental population may not fully capture the complexity of trait variation and may restrict the detection of QTLs present in a broader genetic background [12]. 

The transition from QTL mapping to genome-wide association studies (GWASs) represents a pivotal shift in the field of genetics research. This transition signifies a more comprehensive and high-resolution approach, enabling the detection of subtle genetic variations linked to complex traits. GWASs have emerged as pivotal tools in unravelling the genetic basis of plant disease resistance. GWAS is a powerful way of genetic investigation aiming to identify associations between specific genetic variations, such as single nucleotide polymorphisms (SNPs), and particular traits, such as disease resistance, within a population [13]. Advanced statistical methods are employed to assess the frequency of genetic variations relative to controls to pinpoint variations that are significantly more prevalent in individuals with the trait of interest [14]. By examining genomic datasets, GWASs enable researchers to identify specific genetic variations associated with resistance traits, paving the way for more precise and effective crop breeding strategies to increase disease resistance. Studies have successfully pinpointed genomic regions associated with disease resistance, aiding in the development of resistant crop varieties. For instance, a study highlighted the power of GWAS in identifying genetic loci for resistance to bacterial leaf streak and rice black-streaked dwarf virus (RBSDV) [15,16]. A comprehensive GWAS was conducted on 236 diverse rice accessions, predominantly indica varieties, revealing 12 QTLs across chromosomes 1, 2, 3, 4, 5, 8, 9, and 11 that confer resistance against five distinct Thai isolates of *Xanthomonas oryzae* pv. *oryzicola* (*Xoc*) [15]. Notably, five QTLs exhibited resistance against multiple *Xoc* isolates, among which the *xa5* gene was identified as a potential candidate gene for *qBLS5.1*, while three genes for the pectinesterase inhibitor (*OsPEI*), eukaryotic zinc-binding protein (*OsRAR1*), and NDP epimerase function were proposed as candidate genes for *qBLS2.3* [15]. The identification of these influential genetic factors associated with broad-spectrum resistance potential highlights the significance of GWASs in rice breeding programs targeting BLS resistance. In addition, a study evaluated RBSDV resistance in 1953 rice accessions over three years, revealing lower disease incidences in the Xian/indica (XI) subgroup compared to the Geng/japonica (GJ) subgroup, where a single-locus GWAS, which scrutinized individual variants at specific genomic sites, identified ten genomic regions [16]. Additionally, a multilocus GWAS, which considered multiple genetic variants across the genome, pinpointed five genomic regions linked to RBSDV resistance [16]. From the reported regions, *grRBSDV-6.1* and *grRBSDV-6.3*, haplotype analysis indicated that specific candidate genes, *LOC_Os06g03150* in *grRBSDV-6.1* and *LOC_Os06g31190* in *grRBSDV-6.3*, were associated with resistance differentiation in addition to the three novel resistance regions (*grRBSDV-1.1*, *grRBSDV-7.1*, and *grRBSDV-9.1*) identified [16]. These findings provide valuable insights for breeding RBSDV-resistant rice varieties and serve as a compelling demonstration of the efficacy of GWASs in deciphering the genetic architecture of plant defense mechanisms and identifying pivotal genes and pathways involved in the plant’s response to pathogens.

While QTL mapping is ideal for in-depth studies of a few traits, bulk segregant analysis (BSA) is advantageous for screening multiple traits in large populations, making it a valuable tool in broader genetic studies. BSA is a powerful tool in plant genetics that helps identify genetic markers associated with specific traits, such as disease resistance, with lower cost. This technique accelerates the process of locating genomic regions linked to resistance genes. As described by Majeed et al. in 2022, it is a high-throughput QTL mapping approach that rapidly pinpoints genomic loci regulating a trait of interest, which involves pooling individuals exhibiting extreme trait phenotypes, creating bulks, and then subjecting these bulks to genome-wide analyses [17]. BSA has proven invaluable in various fields, allowing researchers to efficiently unravel the genetic basis of complex traits. Recent advancements in genomic sequencing have given rise to QTL-seq, an innovative next-generation sequencing-based BSA technique. Unlike conventional QTL mapping approaches, QTL-seq offers superior resolution and efficiency in pinpointing genetic markers linked to quantitative traits by leveraging high-throughput sequencing technologies to sequence bulks of individuals with contrasting trait phenotypes, enabling researchers to pinpoint candidate QTLs with remarkable precision [18,19]. It has been instrumental in identifying genomic regions associated with resistance to various diseases in rice, such as bacterial panicle blight (BPB) and dirty panicle disease [20,21]. QTL-seq, in combination with traditional mapping, identified a major QTL for BPB resistance on the upper arm of chromosome 3 containing three genes associated with defense (*OsMADS50*, *OsDEF8*, and *OsCEBiP*) [20]. With the same approach, three QTLs (*qDP1*, *qDP9*, and *qDP10*) were identified to be significantly associated with resistance to dirty panicle disease, which contain genes encoding PR-proteins, subtilisin-like protease, and ankyrin repeat proteins [21]. This approach has gained popularity due to its ability to handle larger populations efficiently and its potential to uncover complex trait variations. Also, it is a rapid and effective approach for identifying genetic loci involved in plant disease resistance, facilitating the development of resistant cultivars.

Furthermore, genome sequencing has transformed our understanding of plant disease resistance, uncovering intricate genetic details that underpin the plant’s ability to fend off pathogens. This technical advancement led to the discovery of various disease resistance genes in other crop systems. For example, Kankanala et al. (2019) delve into the genomics of legume resistance to various plant pathogens [22]. This provides insights into the molecular basis of different levels of host defense observed in both resistant and susceptible interactions by summarizing large-scale genomic studies, shedding light on host genetics changes and enhancing our understanding of plant-pathogen dynamics [22]. This is not only instrumental in identifying existing disease resistance genes, but also in enhancing plant immunity through genome editing technologies like Clustered Regularly Interspaced Short Palindromic Repeats-Cas9 (CRISPR-Cas9). The targeted mutagenesis of genes involved in disease resistance has led to the development of crops resistance to various pathogens, ensuring higher yields and reduced dependence on chemical pesticides [23]. In addition, using comparative genome analysis among different plant species, scientists gain insights into evolutionary aspects of disease resistance genes, aiding in the development of robust, broad-spectrum resistance [24]. This approach is particularly vital in understanding the diverse responses of organisms to pathogens and environmental pressures [25]. As it offers a robust framework for identifying genetic variations linked to resistance, including precise markers and mutations, this approach leads to a refined understanding of resistance dynamics and to the formulation of tailored disease management strategies.

### 2.2. Harnessing Transcriptomics to Safeguard Rice against Disease

Transcriptome profiling involves studying the complete set of RNA transcripts derived from the genome under specific circumstances. During a plant–pathogen interaction, changes in gene expression patterns play a crucial role in defense responses. Various techniques such as RNA sequencing (RNA-seq) and microarray analysis are employed to profile transcriptomes. These technologies enable researchers to quantify gene expression levels, identify alternative splicing events, and detect non-coding RNAs. RNA-seq has been extensively utilized in studying rice diseases to unravel the molecular mechanisms underlying pathogen resistance and susceptibility. Numerous studies have employed RNA-seq to analyze gene expression changes, identify differentially expressed genes, and uncover pathways involved in rice immunity against various pathogens. By utilizing this approach, defense-related genes, such as *PR1b*, transcription factor gene *OsWRKY30*, and *PAL* genes (*OsPAL1* and *OsPAL6*), and pathways like the phenylalanine metabolic pathway, alkaloid biosynthesis pathways (tropane, piperidine, and pyridine), and plant hormone signal transduction pathways were identified from a sheath blight resistant cultivar, suggesting the early activation of an SB-induced defense system [26]. Similarly, a study on the enhanced rice *Xa7*-mediated bacterial blight resistance at high temperature found that the enhanced *Xa7*-mediated resistance at high temperature is not dependent on salicylic acid signaling [27]. A DNA sequence motif similar to known abscisic acid-responsive cis-regulatory elements was also identified in the same study, suggesting that the plant hormone abscisic acid is an important node for crosstalk between plant transcriptional response pathways to high temperature stress and pathogen attack [27].

Transcriptome profiling during plant–pathogen interactions reveals the activation of specific signaling pathways. For example, genes encoding cellular components associated with defense mechanisms, such as pathogenesis-related (PR) proteins, receptor-like kinases (RLKs), and transcription factors, show significant expression changes in the process of a plant–pathogen interaction [28,29]. Transcriptome analysis also unravels complex regulatory networks involved in plant immunity. By identifying key regulatory genes and their targets, scientists can construct intricate networks governing plant defense responses. In addition, like genomics, comparative transcriptomics involves comparing the transcriptomes of different plant varieties, genotypes, or species in response to pathogen attack by highlighting conserved defense mechanisms and revealing unique responses specific to certain plants. A comparative transcriptome analysis revealed that *Rhizoctonia solani* AG1 IA infection activated numerous resistance pathways in rice, involving diverse genes in defense response and signal transduction and highlighting the complex regulation of rice’s pathogen response by multiple gene networks [30]. Also, it revealed the significant activation of metabolic pathways linked to resistance, particularly emphasizing the biosynthesis of jasmonic acid and phenylalanine metabolism [30]. These comparisons enrich our understanding of plant–pathogen coevolution and offer valuable insights for crop breeding programs. This, in return, provides a wealth of information about the molecular mechanisms underlying plant defense responses and aids in deciphering intricate gene regulatory networks and identifying potential targets, thereby offering crucial insights for enhancing crop improvement strategies.

### 2.3. Harnessing Proteomics for Fortifying Rice against Disease

Proteomic studies have also played a pivotal role in unravelling plant immune responses. An analysis of the plant proteome makes it possible to identify key proteins involved in defense pathways and elucidate their functional mechanism. It sheds another light on the complex interactions between plants and pathogens. By comparing protein profiles between infected and uninfected plants, pathogen-responsive proteins have been identified, which include defense-related proteins such as pathogenesis-related (PR) proteins, chitinases, and protease inhibitors [31]. Additionally, proteomics has revealed the modification of host proteins by pathogens to facilitate infection, providing valuable insights into the arms race between plants and pathogens [31].

Techniques such as mass spectrometry (MS) and gel-based methods have been instrumental in characterizing plant defense mechanisms. Advancement in MS and protein isolation techniques has advanced the understanding of subcellular proteomes during plant–pathogen interactions [32]. Proteomic analyses have revealed key proteins pivotal in pathogen recognition, signaling pathways, and metabolic adjustments to combat plant diseases. Notably, receptor-like kinases (RLKs), mitogen-activated protein kinases (MAPKs), and proteins associated with reactive oxygen species (ROS) signaling, hormone modulation, photosynthesis, secondary metabolism, protein degradation, and defense responses have been identified in a rice–*Magnaporthe oryzae* interaction [32]. Furthermore, proteomics has been employed to identify proteins involved in the lesion mimic associated with programmed cell death in rice upon biotic or abiotic stimuli [33]. The study by Yong et al. (2021) revealed several differentially expressed proteins, mainly associated with metabolic and cellular processes, notably including resistance-related proteins such as 14-3-3 proteins, OsPR10, and antioxidases in lesion mimic leaves [33]. This study also elucidated the autoimmunity mechanism in rice [33].

Proteomics has also facilitated the exploration of plant–microbe crosstalk. A recent study delved into the identification and profiling of low-abundant proteins in both compatible and incompatible interactions between rice and *Xanthomonas oryzae* pv. *oryzae* (*Xoo*), utilizing a protamine-sulfate-based method to enrich these proteins, which was followed by their identification and quantification through label-free quantitative proteomics [34]. In incompatible interactions, there was a notable increase in the accumulation of protein kinases, including calcium-dependent protein kinases, PTI1-like tyrosine-protein kinase 1, protein kinase domain-containing protein, and serine/threonine-protein kinase, suggesting their pivotal role in signal transduction for the initiation of immunity in rice [34]. Additionally, mitochondrial arginase-1 encoded by *OsArg1* demonstrated heightened abundance in the incompatible interaction with *Xoo* [34]. The elevated expression of *OsArg1* significantly bolstered rice resistance against *Xoo*, enhancing the expression of defense-related genes such as Chitinase II, Glucanase I, and PR1, which indicates the involvement of this protein in *Xoo* resistance [34]. As a follow-up report, a comprehensive proteome profile was generated elucidating the interaction between rice and *Xoo*, uncovering the proteome changes in the rice cultivars and highlighting the functions of *OsARG1* in plant defense against *Xoo*. [35].

By studying defense responses elicited by bacterial, fungal, and viral pathogens, researchers have unraveled proteins specifically involved in these interactions. Furthermore, comparative proteomics analysis significantly contributes to the study of the intricate molecular mechanisms underlying rice disease resistance. By comparing the protein profiles of susceptible and resistant rice cultivars, researchers have identified various proteins associated with disease resistance pathways, including those involved in signal transduction, defense responses, and metabolic processes [36,37,38]. The proteomics approach also revealed novel insights into the interaction between rice and *Xoo*. In this study, most of the differentially abundant proteins (DAPs) in *Xoo* were related to pathogen virulence, which included the outer member proteins, type III secretion system proteins, TonB-dependent receptors, and transcription activator-like effectors [38]. These DAPs were less abundant in the incompatible interaction and, in this condition, DAPs in rice were mainly involved in secondary metabolic processes, including phenylalanine metabolism and the biosynthesis of flavonoids and phenylpropanoids [38]. This indicates that during incompatible interaction, the rice prevents pathogen invasion and initiates multi-component defense responses.

This knowledge gained from proteomic studies is pivotal for developing strategies to enhance crop resistance against diverse pathogens and provides valuable insights into the dynamic changes occurring at the proteome level during pathogen challenge, shedding light on potential targets for enhancing rice resistance against pathogens. Overall, proteomics serves as a powerful tool for deciphering the molecular basis of rice disease resistance and holds promise for informing strategies to improve technologies for crop protection and agricultural sustainability.

### 2.4. Harnessing Metabolomics for Bolstering Rice Disease Resistance

Metabolomics studies on rice disease resistance involve the comprehensive analysis of metabolites present in different parts of the plant, such as leaves, roots, and seeds, under various conditions. By employing techniques such as liquid chromatography-mass spectrometry (LC-MS), gas chromatography-mass spectrometry (GC-MS), and nuclear magnetic resonance spectroscopy (NMR) to detect and quantify metabolite profiles of diseased and healthy plants, specific metabolites associated with disease resistance can be identified. These metabolites serve as crucial biomarkers, shedding light on the biochemical pathways involved in plant defense mechanisms [39].

Metabolomics studies have successfully linked changes in primary or specialized metabolism to plant defense responses. For instance, the proteomic analysis of *Xoo*-secreted proteins, in vitro and in planta, sheds light on the diverse functions and expression patterns of these proteins during rice bacterial blight infection [40]. The comprehensive proteomic analysis conducted in this study identified 109 unique proteins, elucidating their diverse roles in crucial biological processes such as metabolism, nutrient uptake, pathogenicity, and host defense mechanisms and the observed correlation between protein and transcript abundances unveils the intricate regulatory mechanisms governing protein secretion during in planta infection [40]. In addition, the investigation reveals the potential of transgenic rice expressing these specific secretory proteins to influence cell death signaling, underscoring their pivotal role in pathogenicity [40]. This research significantly advances our understanding of rice bacterial blight disease and provides valuable insights for the development of disease-resistant rice varieties. Furthermore, a study using metabolomics techniques found differences in metabolite accumulation between resistant and susceptible rice plants when exposed to *Xoo* infection [41]. Specifically, plants expressing the *XA21* gene differed from wild-type plants, exhibiting elevated levels of sugar alcohols, tricarboxylic acid cycle (TCA) intermediates, and various other compounds before treatment [41]. Following the inoculation of *Xoo* strain PXO99, *XA21*-expressing plants displayed increased levels of responsive metabolites, such as rutin, pigments, fatty acids, lipids, and arginine, which likely play roles in polyamine biosynthesis and alkaloid metabolism [41]. Additionally, metabolomic analyses have revealed the role of secondary metabolites, such as phenolic compounds and terpenoids, in bolstering plant immunity, which paved the way for developing strategies to enhance the production of disease resistant crops [42]. The same approach was used to identify metabolite levels in rice lines during *Rhizoctonia solani* infection using capillary electrophoresis time-of-flight (CE/TOF)-mass spectrophotometry in positive ion mode where alterations in metabolite levels in inoculated resistant and susceptible rice were examined along the tricarboxylic acid and glycolysis pathways, revealing ten metabolites that were differentially regulated [43]. Notably, chlorogenic acid exhibited increased levels in 32R, a resistant line, while 29S, the susceptible line, pipecolic acid exhibited the highest fold change and significance level and eight amino acids (i.e., glutamate, γ-aminobutyric acid, glycine, histidine, phenylalanine, serine, tryptophan, and tyrosine) displayed elevated levels [43]. These metabolomic signatures often include alterations in the levels of amino acids, organic acids, sugars, and secondary metabolites, which play crucial roles in plant defense mechanisms.

Metabolomics studies have provided valuable insights into the metabolic pathways and key metabolites involved in rice disease resistance. Analyses have revealed the accumulation of defense-related metabolites, such as phenolic compounds, flavonoids, and phytoalexins, in response to pathogen attack. Furthermore, metabolomics approaches have facilitated the identification of metabolic quantitative trait loci (mQTLs) associated with disease resistance, providing valuable targets for breeding programs aimed at developing resistant rice varieties [44]. However, data integration, the standardization of analytical techniques, and the functional validation of identified metabolites are remaining areas for active research [45]. Moving forward, continued advancements in metabolomics technologies and methodologies hold promise for further elucidation of the complex mechanisms underlying rice disease resistance and the acceleration of the development of resilient rice varieties.

## 3. Integrative Omics: Bridging the Layers

Integrative omics approaches merge diverse datasets from genomics, transcriptomics, proteomics, and metabolomics, providing a comprehensive view of biological systems. By integrating these multi-omics data layers, researchers can construct sophisticated systems biology models that unveil complex molecular interactions within organisms. These models offer valuable insights into how genes, proteins, and metabolites interact and function together, facilitating a deeper understanding of biological processes and disease mechanisms.

### 3.1. Integrative Studies in Rice for Disease Resistance

Understanding the crosstalk between different biological components provides a holistic view of rice disease resistance, aiding in the development of resistant rice varieties [46]. By deciphering the genetic basis, gene expression patterns, protein functions, and metabolic changes, these integrative approaches can facilitate precise breeding strategies for improving disease resistance.

By integrating data from genomics, transcriptomics, proteomics, and metabolomics, researchers can perform in-depth analyses, leading to a deeper understanding of plant responses to diseases, and targeted breeding approaches based on the integrative data analysis will accelerate the development of disease-resistant rice varieties. For example, through the combination of GWAS and transcriptional analysis, two genes, *RNG1* and *RNG3* encoding zinc finger protein with a B-box domain and dehydrogenase, respectively, were identified as molecular markers for selecting blast-resistant rice accessions based on their differential expression caused by polymorphisms in 3′-untranslated regions (3′-UTR) [47] (Table 1). Through a combined genetic analysis of leaf blast resistance in upland rice, which included QTL mapping, bulked segregant analysis, and transcriptome sequencing, a novel QTL for blast resistance was fine mapped on chromosome 11 [48]. This study improved the genetic understanding of the mechanism of blast resistance and led to the identification of suitable genotypes with resistance alleles that would be useful genetic resources in rice blast resistance breeding [48] (Table 1). Proteomics study further deepens this kind of molecular genetic knowledge by pinpointing specific proteins involved in the defense mechanisms, while metabolomics complements these findings by identifying metabolites related to disease resistance pathways. For example, molecular interactions between rice and the fungal pathogen *R. solani* could be characterized in regard to gene expression mechanism by employing proteomic and transcriptomic approaches [49]. This led to the elucidation of defense responses in tolerant and susceptible genotypes of *O. sativa* against *R. solani*, which was essential to identify the crucial players of their underlying molecular mechanism [49] (Table 1).

There are also several studies that explored the integration of transcriptomics and metabolomics to understand rice disease resistance. Valuable insights into the defense mechanisms against *Xoo* were obtained from an integrative analysis of metabolomics and transcriptomics data. This analysis revealed the differential expression of several pathogenesis-related genes in *Xoo* PXO99-challenged transgenic *Xa21* plants, such as *GAD*, *PAL*, *ICL1,* and *GS10*, compared to the non-transgenic susceptible parental line [37] (Table 1). Though the *Bph30* gene has been successfully cloned and conferred rice with broad-spectrum resistance to the brown plant hopper (BPH), the molecular mechanisms by which *Bph30* enhances resistance to BPH remain poorly understood [50]. By utilizing both transcriptomic and metabolomic analysis, Shi et al., (2023) elucidated the response of *Bph30* to BPH infestation, using the BPH-susceptible rice cultivar Nipponbare and its *Bph30*-transgenic (BPH30T) line. This study suggests that *Bph30* might coordinate the movement of primary and secondary metabolites and hormones in plants via the shikimate pathway to enhance the resistance of rice to BPH [50] (Table 1). In the case of bacterial panicle blight in rice, by combining QTL-mapping and QTL-seq, one major QTL, *qBPB3.1*, was found in chromosome 3 that conferred resistance to this disease [20].

This integrated approach helps in the precise identification of key molecular players in resistance mechanisms. Integrating omics data also allows the unraveling of regulatory pathways for rice disease resistance by increasing the knowledge of the intricate interactions between genes, proteins, and metabolites, which leads to the identification of regulatory elements and signaling pathways involved in the plant’s defense mechanisms [51]. This knowledge aids in targeted interventions for enhancing disease resistance. Data generated from this integration can be utilized for precision breeding for rice disease resistance and for devising crop management practices. Breeders can utilize these integrated molecular data to develop rice varieties with enhanced disease resistance traits through targeted breeding strategies, resulting in the development of resilient rice cultivars that can withstand various diseases [46]. Furthermore, it can also offer insights into the plant’s response to environmental stressors and diseases, where farmers can adopt informed crop management practices based on these molecular insights, implementing strategies such as tailored irrigation and fertilization practices [52]. Nevertheless, the application of integrative omics for bolstering rice disease resistance has not been explored extensively yet. While research has focused more on abiotic stress tolerance and agronomic traits, there is a notable gap in leveraging these approaches for enhancing disease resistance in rice. Closing this gap could unlock novel insights into the molecular mechanisms underlying disease resistance and lead to the development of resilient rice varieties capable of withstanding diverse biotic challenges.

### 3.2. Integrative Studies in Rice for Tolerance to Abiotic Stresses

This integrative approach has also been transforming the understanding of rice abiotic stress tolerance, leading to remarkable advancements in crop improvement strategies. Several studies exemplify the practical applications of these techniques, showcasing their effectiveness in enhancing rice resilience to various environmental stresses (Table 1). The integration of multi-omics data has been instrumental in deciphering the complex interactions between genes, proteins, and metabolites upon salt stress. A better understanding of salt tolerance mechanisms through the integration of genomics, transcriptomics, and other omics information can accelerate rice breeding for developing salinity tolerant rice varieties [53]. Comprehensive multi-omics approaches also have been instrumental in understanding the components of drought tolerance in rice. Comparative mapping within and across species has provided a holistic view of the genetic factors contributing to drought resistance [54] (Table 1), and key transcription factors for allantoin biosynthesis, *OsERF059* and *ONAC007*, were found to be important for enhancing drought tolerance in rice through combining transcriptomic and metabolic analyses [55]. For heat tolerance, integrating GWAS and transcriptomics has mapped the candidate locus, *LOC_Os07g48710*, responsible for heat tolerance in rice, which encodes a VQ domain containing protein, providing valuable genetic markers for molecular breeding programs for the enhancement of heat resilience in rice varieties [56] (Table 1). In terms of metal toxicity, combined transcriptomics and metabolomics have provided a comprehensive view of the molecular processes underlying the arsenic stress response by identifying arsenic-responsive genes and metabolites, which aids in our understanding of the intricate pathways involved in detoxification and tolerance mechanisms [57] (Table 1). For cold tolerance, the integration of transcriptomic and metabolic profiling revealed that *OsSEH1,* encoding a nucleoporin/WD40 domain-containing protein, plays a role in the oxidation–reduction process contributing to cold tolerance in rice [58] (Table 1).

Integrative omics approaches have become increasingly prevalent in enhancing rice tolerance to abiotic stressors, such as salinity and drought, resulting in significant agricultural advancements and offering a comprehensive understanding of stress response mechanisms, aiding in the development of stress-tolerant rice cultivars.

### 3.3. Navigating the Path to Precision by Combinining QTL Mapping and Omics

Combining mapping and omics data offers a holistic understanding of the genetic basis and molecular pathways governing disease resistance in rice. By overlaying genetic maps with omics datasets, researchers can prioritize candidate genes and regulatory pathways for targeted breeding efforts. The fusion of traditional genetic mapping techniques with cutting-edge omics technologies offers unprecedented insights into the genetic architecture underlying disease resistance traits in rice [59]. Because genomic studies elucidate genetic variations, including SNPs, insertions, deletions, and structural variants, across diverse rice germplasms, researchers can effectively correlate genetic variants identified through mapping studies with changes in gene expression, protein levels, and metabolite concentrations revealed by omics analyses [60]. This correlation provides valuable insights into the functional implications of genetic variations and their roles in disease resistance mechanisms [59]. By correlating rice genetic variants with gene expression data obtained through transcriptomics, researchers can pinpoint genes that exhibit differential expression patterns in response to disease challenges and this correlation enables the identification of key genes involved in the rice plant’s response to disease, providing insights into its defense mechanisms and potential targets for genetic improvement strategies. This allows for the identification of candidate genes associated with disease resistance. By comparing protein profiles between resistant and susceptible rice varieties and associating them with genetic variants, researchers can pinpoint proteins crucial for disease resistance pathways. This approach allows for the identification of key proteins involved in the defense mechanisms of rice against diseases. Furthermore, researchers can identify biomarkers associated with disease resistance and understand the involved metabolic pathways by analyzing metabolite concentrations resulting from the presence of genetics variants. By comprehensively studying metabolite profiles and their variations, researchers gain insights into the underlying mechanisms of disease resistance and can develop strategies for enhancing crop resilience.

In summary, integrating QTL mapping with omics allows for the identification of specific genes, proteins, and metabolites that play crucial roles in combating diseases, leading to the development of more resilient rice varieties. By doing so, researchers gain a deeper understanding of the genetic basis of disease resistance and identify key biomarkers and pathways involved in the resistance mechanisms [59,60]. This integrated approach enhances the precision of breeding efforts by providing insights into the functional roles of disease-related biomarkers and their interactions, ultimately resulting in the selection of superior rice genotypes with enhanced disease resistance.

**Table 1 plants-13-01205-t001:** List of studies utilizing integrative mapping and omics to identify and characterize rice resilience to biotic and abiotic challenges.

Abiotic/Biotic Factors	QTL/Loci/Genes (Gene Products)	Methods	Critical Information	Reference
Drought tolerance	*OsERF059* (ethylene response factor 59) and *ONAC007* (NAC domain-containing protein 7)	Transcriptomics and metabolomics	⮚ Application of allantoin increases the expression of the ROS scavenging genes enhancing drought tolerance ⮚ These ROS scavenging genes are major components for drought tolerance	Lu et al. (2022) [55]
Heat tolerance	*qHT7*/*LOC_Os07g48710* (VQ motif-containing protein 30)	GWAS and transcriptomics	⮚ *OsVQ30* as component for heat tolerance ⮚ Identification of accessions (Geng/Jap, Hap4) as possible source of heat tolerance	Li et al. (2023) [56]
Arsenic toxicity	DEGs and DAMs ion transporters, ROS, etc.	Transcriptomics and metabolomics	⮚ Identification of DAM-related genes such as dermatan L-iduronate for As tolerance ⮚ Biomarkers for As tolerance	Ma et al. (2023) [57]
Cold tolerance	*OsSEH1* (nucleoporin SEH1)	Transcriptomics and metabolomics	⮚ Identification of *OsSEH1* as positive regulator for cold tolerance ⮚ Identification of redox genes regulated by *OsSEH1* ⮚ regulation of ABA via *OsSEH1* for cold tolerance	Gu et al. (2023) [58]
Blast resistance (*Magnaporthe grisea*)	*RNG1* (Zinc finger protein with B-box-domain) and *RNG3* (Dehydrogenase)	GWAS and transcriptomics	⮚ Knocking out *RNG1* and *RNG3* genes enhanced resistance to rice blast ⮚ Markers for identifying rice-blast resistant lines	Xu et al. (2023) [47]
Blast resistance (*Magnaporthe grisea*)	*Os11g0700900* (glycoside hydrolase), *Os11g0704000* (SelT selenoprotein family), *Os11g0702400* (zinc finger, C2H2-type domain containing protein), and *Os11g0703600* (hypothetical protein)	BSA, QTL-mapping, and transcriptomics	⮚ A novel fine-mapped QTL in chr.11 for blast ⮚ Genes and variants within this QTL that could serve as biomarkers and characterization	Tan et al. (2022) [48]
Sheath blight resistance (*Rhizoctonia solani*)	*LOC_Os12g44010.1* (purple acid phosphatase 10b)*, LOC_Os04g43290.3* (actin-related protein (ARP) C2 subunit)*, LOC_Os11g48000.1* (EPF zinc-finger)*, LOC_Os09g29480.2* (2-aminoethanethiol dioxygenase)*, LOC_Os06g45890.1* (MYB-like transcription factor)*, LOC_Os04g46980.1* (cis-zeatin-O-glucosyltransferase)*,* and *LOC_Os09g12790.1* (potassium channel protein)	Proteomics and transcriptomics	⮚ Identification of important genes associated with sheath blight resistant line that could be utilized for characterization ⮚ Identified microRNAs and genes useful to develop markers for selection	Prathi et al. (2018) [49]
Bacterial blight resistance (*Xanthomonas oryzae* pv *oryzae*)	*GAD* (Glutamate decarboxylase)*, PAL* (Phenylalanine ammonia-lyase)*, ICL1* (Isocitrate lyase), and *GS10* (Glutathione-S-transferase)	Transcriptomics and metabolomics	⮚ Identification of specific receptor–effector interaction and downstream pathways that could be used for specific gene targeting	Sana et al. (2010) [41]
Brown plant hopper resistance (Tungro virus)	*Bph30* (Leucine rich repeat (LRR) family protein)	Transcriptomics and metabolomics	⮚ Identification of genes associated with primary and secondary metabolites and hormones in plants via the shikimate pathway for gene targeting enhancing BPH resistance	Shi et al. (2023) [50]
Bacterial panicle blight resistance (*Burkholderia glumae*)	*qBPB3.1*	QTL-mapping and QTL-seq	⮚ Identification of a qBPB3.1 (19.82%, 5.49–26.24%) conferring resistance to both BPB and SB ⮚ qBPB3.1 markers for selection ⮚ qBPB- relate genes for further characterization	Ontoy et al. (2023) [20]

## 4. Challenges and Gaps: Exploring the Intersections

While both genetic mapping and omics studies have significantly contributed to the field of rice disease resistance, integrating these approaches faces several challenges and gaps that need to be resolved for an efficient and precise characterization of rice resistance against diseases and its application to rice breeding for enhancing disease resistance through the development of more reliable molecular markers. One major concern is the integration of diverse data types generated from diverse platforms. Genomic data provide information about genetic loci, while transcriptomic data reveal gene expression patterns. Integrating these diverse datasets requires robust computational methods that can align and analyze data from various sources [61].

The accuracy of the functional annotation of genomic variants is another critical issue. Understanding the functional significance of genetic variants detected through mapping studies is essential, and integrating omics data can aid in annotating these variants by correlating them with gene expression profiles and protein functions [62]. This integration is crucial for deciphering the molecular basis of disease resistance. Typically, stress often induces dynamic changes in gene expression, protein abundance, and metabolite levels, so static omics snapshots might miss these dynamic responses [63]. In this context, incorporating time-series omics data during disease progression or dynamic omics profiling is essential to capture the temporal aspects of the host–pathogen interaction. Integrating networks derived from mapping and omics data can provide insights into key regulatory nodes, but existing tools lack robustness in handling large-scale integrated networks [46]. Advanced network analysis algorithms are required to identify central genes, proteins, and metabolites crucial for disease resistance.

Lastly, environmental factors significantly influence disease resistance. The integration of environmental parameters like temperature, humidity, and soil quality into multi-omics datasets allows researchers to identify genes and pathways modulated by environmental cues. Understanding genotype–environment interactions can provide valuable insights into the adaptability of rice varieties under varying environmental conditions [64]. As the current global climate change greatly affects rice diseases, through compromised disease resistance, increased pathogen populations, or the emergence of new pathogens [65], the traits in resilience to abiotic stresses, especially heat stress, should be an essential part in the integrated multi-omics studies and breeding programs of rice for better understanding and enhancing disease resistance.

Addressing the research gaps related to data integration, functional annotation, dynamic profiling, network analysis, and environmental interactions is crucial. Future research should focus on refining computational models, enhancing data integration methods, and exploring emerging omics technologies to further unravel the complexities of rice disease resistance [53]. By overcoming these challenges, we can unravel the intricate molecular mechanisms governing disease resistance with precision and efficiency, paving the way for the development of disease-resistant rice varieties.

## 5. Conclusions

The intersection of traditional genetic mapping and cutting-edge omics technologies has opened new horizons in understanding and enhancing rice disease resistance. The integration of mapping and omics data is not merely a combination of techniques; it represents a paradigm shift in our approach to studying complex biological systems. Regarding the significant advancements, challenges, and prospects of this integration reviewed in this article, it becomes evident that this interdisciplinary approach offers unprecedented precision in characterizing rice disease resistance. Integrative studies encompassing these technologies hold promise in not only characterizing resistance but also in engineering highly resilient rice varieties. This represents a transformative approach in the realm of rice disease resistance research and breeding. Furthermore, the knowledge gained from these studies will have a huge impact on the development of innovative strategies for sustainable agriculture.

## Figures and Tables

**Figure 1 plants-13-01205-f001:**
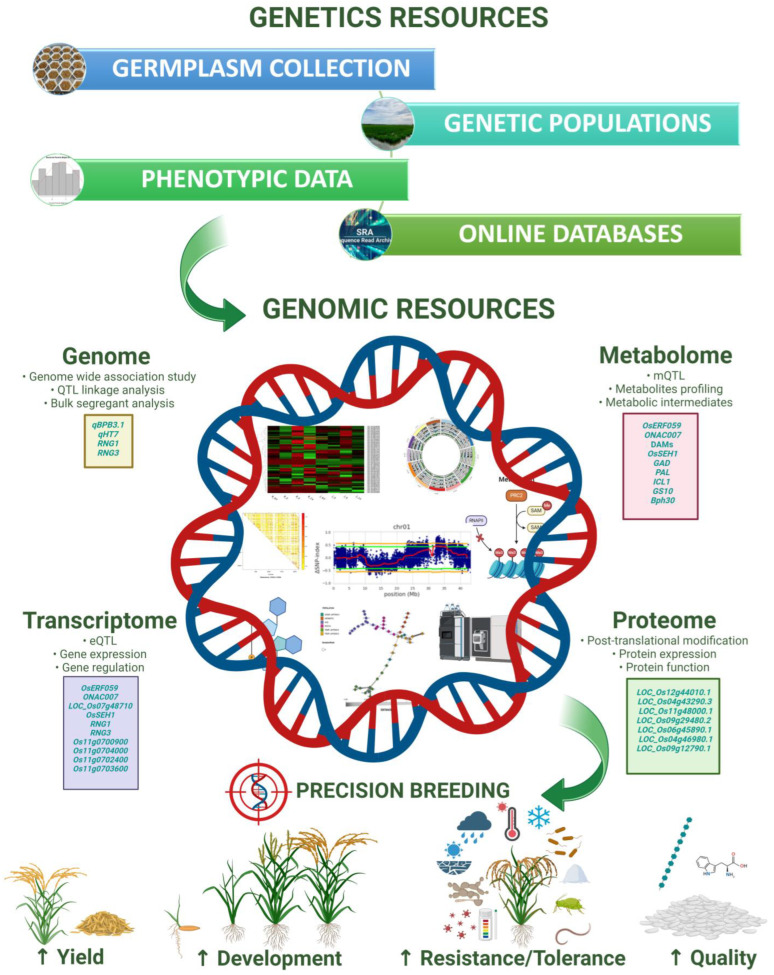
Utilizing genetic and genomic resources for rice breeding enables targeted selection of traits, such as disease resistance, yield potential, quality, and abiotic stress tolerance, leading to the development of improved rice varieties with enhanced productivity and resilience. Some images in this figure are from ‘BioRender.com’.

## Data Availability

Not applicable.

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
