# Peer review of "Mapping and Omics Integration: Towards Precise Rice Disease Resistance Breeding"

_plants, 2024, doi:10.3390/plants13091205_

Round 1

Reviewer 1 Report

Comments and Suggestions for Authors

In this manuscript (plants-2972919) entitled "Mapping and Omics Integration: Towards Precise Rice Disease Resistance Breeding" submitted to Plants, John Christian Ontoy and Jong Hyun Ham have discussed integration of mapping and omics data on rice breeding for enhancing disease resistance. This review is interesting and well-written, but the current version of this manuscript needs to be revised for publication.

Major points:

1. For the Figure 1, QTLs discussed in this review should be included in this revised Figure.

2.For the Table 1, Contribution of QTLs to rice stress resistance and evidence should be included in this revised Table.

3, Authors should consider to add a new section to discuss perspectives on integration of mapping and omics data on rice breeding for enhancing disease resistance in the revision.

4, Line number should be shown in the revised manuscript.

Minor points:

1. Full names of abbreviations like QTL, R, and CRISPR-Cas9 should be spelt out at their first appearance. Authors should check all abbreviations employed in the manuscript.

2. Authors need to standardize references according to the Plants template. For instance, “Int J Mol Sci.” should be replaced with “Int. J. Mol. Sci.” (Reference 1).

Reviewer 2 Report

Comments and Suggestions for Authors

Please see the attached MS for my comments.

Comments on the Quality of English Language

Please see the attached MS for my comments on the quality of the English language.  

Round 2

Reviewer 1 Report

Comments and Suggestions for Authors

Authors have addressed my concerns in the revision